# Stratifying Cumulus Cell Samples Based on Molecular Profiling to Help Resolve Biomarker Discrepancies and to Predict Oocyte Developmental Competence

**DOI:** 10.3390/ijms22126377

**Published:** 2021-06-15

**Authors:** Osman El-Maarri, Muhammad Ahmer Jamil, Maria Köster, Nicole Nüsgen, Johannes Oldenburg, Markus Montag, Hans van der Ven, Katrin van der Ven

**Affiliations:** 1Institute of Experimental Haematology and Transfusion Medicine, University of Bonn, Sigmund-Freud Str. 25, 53127 Bonn, Germany; Muhammad.Jamil@ukbonn.de (M.A.J.); Nicole.Nuesgen@ukbonn.de (N.N.); Johannes.Oldenburg@ukbonn.de (J.O.); 2Department of Gynecologic Endocrinology and Reproductive Medicine, University Women’s Hospital, 53127 Bonn, Germany; labor@kinderwunschzentrum-bonnerbogen.de (M.K.); mmontag@ilabcomm.com (M.M.); bonnhv@gmail.com (H.v.d.V.); 3Kinderwunschzentrum Bonner Bogen, Joseph-Schumpeter-Allee 1, 53227 Bonn, Germany; 4Ilabcomm GmbH, Eisenachstr. 34, 53757 St. Augustin, Germany; 5Zentrum für Gynäkologische Endokrinologie und Reproducktionsmedizin Bonn, Godesberger Allee 64, 53175 Bonn, Germany

**Keywords:** ART, ICSI, cumulus cell, mRNA expression profiling, CpG methylation profiling, oocyte competence, epigenetic biomarker, FSH, pathways analysis, upstream regulators

## Abstract

To increase the efficiency of assisted reproductive techniques (ART), molecular studies have been performed to identify the best predictive biomarkers for selecting the most suitable germ cells for fertilization and the best embryo for intra-uterine transfer. However, across different studies, no universal markers have been found. In this study, we addressed this issue by generating gene expression and CpG methylation profiles of outer cumulus cells obtained during intra-cytoplasmic sperm injection (ICSI). We also studied the association of the generated genomic data with the clinical parameters (spindle presence, zona pellucida birefringence, pronuclear pattern, estrogen level, endometrium size and lead follicle size) and the pregnancy result. Our data highlighted the presence of several parameters that affect analysis, such as inter-individual differences, inter-treatment differences, and, above all, specific treatment protocol differences. When comparing the pregnancy outcome following the long protocol (GnRH agonist) of ovarian stimulation, we identified the single gene markers (*NME6* and *ASAP1*, FDR < 5%) which were also correlated with endometrium size, upstream regulators (e.g., *EIF2AK3*, *FSH*, *ATF4*, *MKNK1*, and *TP53*) and several bio-functions related to cell death (apoptosis) and cellular growth and proliferation. In conclusion, our study highlighted the need to stratify samples that are very heterogeneous and to use pathway analysis as a more reliable and universal method for identifying markers that can predict oocyte development potential.

## 1. Introduction

Infertility is a global concern affecting an estimated 10% of reproductive-age couples [1]. When sperm motility, morphology, or counts are affected, intracytoplasmic sperm injection (ICSI) must be used [2]. Despite technological advances, successful live birth rates are far from 100%. Several markers/parameters are investigated to monitor the entire process and to select the best embryos for intra-uterine transfer to increase the rate of live births. These potential markers are linked to the biological process of follicular and oocyte development, maturation, and ovulation and include a variety of types: (A) endometrium wall thickness and volume [3], (B) hormone levels [4], (C) follicle number and size [5], (D) gene expression in granulosa [6] or mural granulosa cells [7], (E) gene expression in cumulus cells [6,7,8,9,10,11,12,13,14,15,16,17,18,19,20,21,22,23,24,25,26,27], (F) microRNA expression in cumulus cells [28], (G) gene expression in follicular fluid cells [29], (H) zona pellucida morphology parameters [30], (I) analysis of polar body chromosome number and structure [31], and (J) morphological appearance of the oocyte [32].

Of the above-mentioned approaches, gene expression analysis (categories D to G) offers a promising methodology for monitoring oocyte development due to its relative simplicity. However, different studies have failed to identify universal markers linked with good developmental potentials (Appendix A). This is possibly due to the heterogeneity of samples and to differences in their preparation because sample sources could be mural cells, cumulus cells (inner or outer), granulosa cells (precursor of mural and cumulus cells), or follicular fluid derived cells. The heterogeneity of samples in biological developmental stage and purity of sample preparation, further contributes to the inconsistency of results. Moreover, the degree of maturity of the retrieved oocyte has been found to be associated with a specific expression signature that could further cause discrepancies when comparing results between different studies [33,34]. Likewise, differences in methodology also contribute to the variability of results, as a number of studies have applied broad genomic-based approaches to identify genome-wide expression using microarrays or RNA-seq, while others have focused on the expression of a limited number of genes related to specific pathway(s) that were shown to play important roles in oogenesis. Additionally, different treatment protocols further increase the variability of results.

Several studies performed controlled experiments to study the effect of parameters/factors that could introduce systematic bias into the results. Indeed, age [35], stimulated in vitro fertilization cycles rather than natural cycles [36], differences in treatment protocols, as well as medications used during ovarian stimulation [37,38,39] were seen to have an effect. Additionally, a patient’s genetic constituents also affect the clinical output and success of treatment. For example, the influence of polymorphism in the follicle stimulating hormone receptor (FSHR) and the luteinizing hormone (LH) beta subunit [40] and alternative skipping of exon 2 or 3 in the FSHR [41] have been reported to affect the ovarian responses to the FSH.

Therefore, it is expected that only an exact comparison among similar samples can identify reliable biomarkers that can correlate with the clinical outcome. For this reason, we revisited this theme by analyzing well-controlled samples and stratifying these into groups according to their molecular signature and known differences in biological and experimental conditions. This was possible because of our unique experimental design, which includes: (1) similar age group of patients, (2) comparable male factor infertility severity defect, and (3) the analysis of multiple non-pooled cumulus-cell clusters from each donor woman corresponding to individual oocytes being used for simultaneous transfer. This study design enabled comparisons of single patients with each other and prediction of the relative activation status of specific biological pathways. In addition, we compared the generated genomic data to the available clinical parameters: spindle presence, Zona pellucida birefringence, pronuclear pattern, estrogen level, endometrium size, and lead follicle size. Our results re-emphasized that the sample type, treatment protocol, and individual differences can influence gene expression and methylation profiling. Additionally, we showed that pathway analysis is providing better parameters for predicting clinical outcomes. We also provide, for the first time, evidence that pregnancy-negative associated samples are classified based on upstream regulators (like FSH) into two negative groups that could either be over- or understimulated in comparison to positive samples. Finally, a correlation between gene expression related to growth and apoptosis and clinical parameters like the level of estrogen, provided assurance of the accuracy and relevance of the results.

## 2. Results

### 2.1. Clinical Characteristics of the Collected Samples 

The summary of samples characteristics is shown in Appendix A. For this study, we collected the outer cumulus cells only by mechanical separation from the inner cumulus directly surrounding the oocyte, and that needed to be done by enzymatic treatment. The latter could introduce variability to the cumulus cells molecular markers while the former could be obtained without introducing molecular changes. We grouped the obtained samples into three subgroups: (1) short protocol negative (there are no positive samples for the short protocol), (2) long protocol positive, and (3) long protocol negative. We plotted response-fitting curves representing the main clinical data (estrogen levels, endometrium thickness, and maximum follicle size) and observed clear differences among the three groups (Appendix A). We then compared the day-12 responses (predicted from fitting curves if not available) and the curve slope for all the sample groups to reflect the personal/individual response of the treatment. A clear difference was observed in the long positive protocol samples, which showed a clear tendency for a higher, faster response than the long protocol negative samples (Appendix A).

Appendix A shows the correlations among the available clinical parameters. When considering the samples generated using the long protocol (right panel) we found a total of 11 significance correlations out of a possible 55. The top significant correlations (excluding those that involve slopes) were between estrogen level and lead follicle at day 12 (*p* = 1.51 × 10^−4^; Pear. Corr. = 0.776). Followed by endometrium size and estrogen levels at day 12 (*p* = 0.014; Pear. Corr. = 0.57), followed by pronuclear pattern and estrogen levels at day 12 (*p* = 0.022; Pear. Corr. = −0.535). Additionally, of interest is the observed correlation between age and endometrium size at day 12 (*p* = 0.019; Pear. Corr. = 0.547). 

### 2.2. Correlation of Clinical Parameters with Both Genome Wide Expression and DNA Methylation Data

We investigated the correlations between the gene expression/DNA methylation data and the recorded clinical parameters. First, we considered all samples including both the long and the short protocol. At *p* = 0.01, we observed a number of correlated gene probes ranging from 48 with spindle presence or absence up to 1119 with lead follicle size (Appendix A). A significant number of overlapping correlated genes were observed between estrogen levels, endometrium size, and lead follicle size (Appendix A). When considering the long protocol samples, only the range of correlations between the clinical parameters and the gene expressions ranged from 16 probes/genes for spindle presence to 847 for estrogen levels (Appendix A). However, gene overlap correlations with different clinical parameters were considerably fewer; however, 38 and 233 overlaps were still observed between estrogen levels and endometrium size and lead follicle size, respectively (Appendix A). The groups of correlated genes in all cases were highly enriched with cancer-related genes, thus much related to growth and the cell cycle (Appendix A). 

Similarly, we determined the methylation at CpG sites that correlated with clinical parameters and observed a similar pattern to the correlation of gene expression and clinical parameters (Appendix A). For example, the highest number of correlated CpG methylations or gene expressions with clinical parameters were for estrogen levels and endometrium size, while the lowest numbers for both expression and methylation correlated with spindle presence and zona pellucida birefringence (for long protocol only samples, correlation between the number of correlated probes and correlated genes = 0.93, *p* = 0.0003). However, when comparing CpG sites and genes we observed little overlap for the same clinical parameters ranging from 10 to 22%.

### 2.3. Inter-Individual Differences Order/Separate Samples According to Donor Identity Showing High Significance in Genes Related to Protein Metabolism

As a first step after generating expression and methylation profiles of the individual non-pooled 24 cumulus samples, we investigated whether we could recognize the samples and group them based on their original female donor, i.e., if there were an inter-individual signature. Unsupervised clustering illustrated in heatmaps and 3D-PCA plots showed this to be the case (Figure 1A). In fact, the expression data clearly demonstrated that the cumulus clusters samples from the same donor were grouped together and were closer to each other than the samples from other donors (represented by the nearest neighboring lines connecting the spheres in Figure 1A, upper left, and the unsupervised hierarchical clustering on the heatmaps, lower left). What emphasized the inter-individual signature effect and excluded the effect of experimental procedure(s) was the fact that the samples were derived from two different treatments but from the same donor (i.e., donated at different hormone stimulation cycles) and were still grouped closely together. This held true for donor 1 (eight cumulus samples from two treatments: blue annotation) and donor 2 (four cumulus samples from two treatments: pink annotation). The latter results emphasized the accurate experimental procedures that maintained individual sample-specific information. 

While correlation analysis between expression and methylation showed negative as well as positive correlations, at the transcription start site (TSS) a clear tendency was found for negative over positive correlations (Figure 1B). In total, 1043 CpG methylations overlapped with 659 annotated genes. The highest positive correlation was found at GAA (with cg16464924: *p* = 0.84 × 10^−6^; *Rho* = 0.88), while the lowest negative correlation was seen at C2Oorf24 (with cg19216162: *p* = 1.5 × 10^−6^; *Rho* = −0.88) (Figure 1B). 

Supervised clustering by applying a false discovery rate (FDR) of 5% identified 4358 expression probes (corresponding to 3869 genes) and 10,453 CpGs to clearly distinguish the 24 samples based on female donor origin (Figure 1A). Gene ontology analysis on genes showing inter-individual differences in expression indicated that ontological terms related to the following three categories were the most common. The first, cellular metabolism, included metabolic, biosynthetic, organic acid metabolic, lipid, and catabolic processes. The second, protein and transport, concerned protein localization, protein complexes, and transport endosomes. The last category, cellular regulation, included cell death, the cellular response stimulus, and the negative regulation process (Figure 1C, left panel). The variably expressed genes showed inter-individual difference codes for proteins localized in the intracellular organelle, protein complexes, Golgi apparatus, endoplasmic reticulum, mitochondrial membrane envelope, and matrix (Figure 1C, right panel). The cellular localization component emphasized the previous biological processes related to protein metabolism and transport.

Furthermore, to obtain more detailed information on the group of variably expressed genes and their biofunctions, we performed genomic analysis based on Ingenuity Pathways Analysis (IPA). The IPA database uses experimental information on protein interaction and function from the published scientific literature. This analysis can reveal the affected pathways, the regulators that triggered the majority of changes and what could be predicted from these; thus, it can show what kind of disease these could lead to. 

Canonical pathway: The top-10 enriched canonical pathway analyses showed that metabolic pathways are mostly affected, and this includes protein synthesis and several signaling pathways in the categories of cellular growth, proliferation, and development (Figure 1D).

Upstream regulator: In this analysis, the IPA database searches for molecules or proteins, such as hormones and transcription factors, that can influence the expression of genes in the given gene sets (i.e., in genes showing significant variability). Based on the set of variable gene expression, this analysis predicted that FSH and LH (two main hormones used in the treatment) would have the top overlap *p*-value between the total possible number of genes regulated by these hormones and the expected genes regulated by these hormones in the dataset (Figure 1E). This finding reflected a gene expression signature of a possible quantitative effect of the drugs used either regarding the actual dose effect in relevant tissue or the metabolic effect influenced by genetic polymorphism. 

Disease and function: The third analysis includes calculating the occurrence of variable genes in known diseases, physiological functions, or biofunctions. (Figure 1F). Here, 7 of the top-10 significant functions fell into the category of cell death and survival, while subcategories included apoptosis, necrosis, and proliferation.

### 2.4. Differences between Two Treatments with the Same Protocol and the Same Individual

One of the donors (number 1 in Appendix A) underwent two successive treatments leading to two sample groups, VS1 (first treatment) and VS6 (second treatment). The second treatment led to a pregnancy and delivery. As both samples were derived from the same individual this provided the opportunity to study expression differences between the two treatments on an identical genetic background. Such expression differences could be responsible for, or at least contribute to, the pregnancy outcome. In comparison to VS6 (positive pregnancy), there were 192 differential expression genes with 122 overexpressed and 70 under expressed in VS1 (negative pregnancy) (Appendix A). General ontological analysis showed that the affected biological processes included overexpression in VS1 (negative pregnancy) of the metabolic process, regulation of the metabolic process, the L-serine metabolic process, and proline transport. Most of these molecules were localized in intracellular organelles, the mitochondrial envelope, and the nuclear lumen (Appendix A). Detailed canonical pathway analysis based on the IPA showed that the top-10 affected pathways included some regulatory/metabolic, cancer, and autophagy pathways (Appendix A). Coordinated upstream regulators were also detected, suggesting a common regulatory mechanism for affected molecules, possibly excluding random stochastic effect(s) (Appendix A). Disease and biofunction grouping of the affected molecules via IPA revealed the top categories related to cell death and gene expressions that were predicted to be downregulated in the negative VS1 samples (Appendix A).

The above-mentioned analysis performed on genetically identical samples provided a good opportunity to analyze differential expressions that could be directly linked to a clinical outcome. In fact, the differences highlighted metabolic process, cancer-related pathways, and apoptosis, revealing the different cellular state found in the two groups. The highest differentially expressed molecule was FOSB (higher in VS6), an AP-1 transcription factor subunit involved in cellular proliferation as well as the cellular response to calcium ions, hormone stimulus, cAMP, corticosterone stimulus, mechanical stimulus, morphine, progesterone stimulus, female pregnancy, and regulation of transcription from RNA polymerase II promoter). Thus, it appears that the VS1 cells were isolated from less proliferative cells that did not favor complete oocyte maturation. Therefore, these cells had either not yet reached, or had passed, their optimal developmental state. These differences between the VS1 and the VS6 samples could be induced by differences in hormone treatment (including human errors of treatment regime compliance) or differences in treatment duration and timing of sample collection.

### 2.5. Differences between Long and Short Stimulation Protocols

Of the eight donors, three were treated according to the short protocol and five according to the long protocol. As there are basic differences between both protocols regarding duration and the type of drugs or hormones used, we compared both sample groups for major differences. Unsupervised hierarchical clustering of the samples (heatmap and 3D-PCA plots, Appendix A) using the expression data revealed good separation between the two groups, suggesting that we can discriminate between the two stimulation protocols samples based on gene expression profile. Similar results were obtained using CpG methylation data. When applying an FDR < 5%, we obtained 242 differential expression probes (corresponding to 220 genes) and 83 differential methylated CpGs (Appendix A). In this comparison, general gene ontology analysis also revealed biological processes related to metabolic and growth regulation. This was further emphasized by their localization in internal vesicle organelles, such as the endoplasmic reticulum, Golgi apparatus, and mitochondria (Appendix A). Detailed IPA analysis showed canonical pathways that mainly involve metabolic pathways (Appendix A). Some upstream regulators were also identified but without reaching an activation Z score above 2 (Appendix A). In the disease and function category, when taking the Z score and activity prediction into consideration, a clear picture emerged: categories related to cell death were activated, while cellular growth (in development and function) was predicted to be inhibited in the long protocol compared to the short protocol (Appendix A).

### 2.6. Molecular Profile of Individual Non-Pooled Cumulus Cells Derived from Positive or Negative Pregnancy Test (Long Stimulation Protocol Only)

Taking into consideration the result of the previous section, we next analyzed the samples derived via the long protocol only. While analysis of all data from the long and the short protocols revealed no significant differences, by stratifying the samples according to the stimulation protocol, we obtained a significant number of differentially expressed genes from the long protocol samples (Figure 2). Non-supervised hierarchical clustering did not reveal clear grouping according to pregnancy outcome. However, 3D-PCA plots showed that the positive samples were largely aligned according to PCA component 3, yet without suggesting the alignment reason. Moreover, supervised clustering at *p* = 0.05 revealed 786 and 771 under- and overexpressed probes, respectively, in positive samples. The most significant results were obtained at FDR < 0.05 with three probes corresponding to genes *NME6* and *ASAP1*, which were underexpressed in positive samples (Figure 2A, left panel). These two genes were also found to be top correlated with two of the clinical parameters: endometrium size and lead follicle size slope (Appendix A). More than 23,000 different CpGs were found to be differentially methylated at *p* = 0.05, but there was no significance at FDR < 0.05 (Figure 2A, right panel). The top 10 significant CpG sites are shown at Appendix A. We next studied the correlation between the expression of differentially expressed genes and their underlying methylation in a region covering 10 Kb upstream and 10 Kb downstream of the TSS. Here, we also found an expected surplus of negative correlations, especially at the TSS (Figure 2B).

In a further step to validate the relevance of differentially expressed genes between the negative and positive samples, we analyzed the intersection between differentially expressed genes in a negative/positive and a VS1/VS6 comparison. As the latter were derived from the same individual/donor, they served as a control since differences were more likely to reflect treatment without genetic interference. We hypothesized that if the list of differentially expressed genes were relevant to the pregnancy outcome then (1) the under- or overexpressed genes in both comparison groups (negative/positive and VS1/VS6) would have considerable overlap and (2) no or little interactions should be observed between overexpression in one comparison (e.g., negative/positive) and underexpressed in the second (e.g., VS1/VS6), and vice versa. Indeed, we observed the expected clear phasing, and the results are illustrated in Figure 2C.

Ontology analysis on differentially expressed genes showed overexpression in genes related to negative regulation (cell death activation in the negative group) in negative samples. Genes related to protein localization were relatively overexpressed in positive samples (Figure 2D). A detailed IPA analysis showed affected canonical pathways including metabolic processes but also genes in mismatch repair that were underexpressed in positive samples (Figure 2E). Upstream regulators listed FSH as the second from top in the list with an overlap *p*-value of 8.76 × 10^−7^ and a predicted inhibition Z score (in positive samples) of −0.724 (Figure 2F). Disease and biofunction distribution of the differentially expressed genes revealed activated Z scores related to cellular proliferation and negative Z scores to cell death (Figure 2G).

### 2.7. Detailed One to One Comparison between Positive and Negative Samples

In the previous section, we showed that the positive samples (designated here as the red-positive group) settled themselves, as seen by 3D-PCA plot, between two groups of negative samples designated here as blue-negative and yellow-negative groups (Figure 3A). In fact, analyzing these three groups for differences using ANOVA (at FDR < 5%) revealed that they could easily be separated based on 739 expression probes (Figure 3B); the same grouping also showed up on a PCA plot when considering differential methylation data of >10% between positive and negative samples (Appendix A). To clarify the reason behind this arrangement we looked for systematic differences among the three groups and performed IPA Z score comparisons for upstream regulators, canonical pathways, diseases, and biological function (Figure 3C–E) between each sample group. Two samples corresponded to one patient from the positive group (middle red-positive in Figure 3A), and each sample was compared to one from the negative group (upper blue-negative group and lower yellow-negative group in Figure 3A). The term sample here refers to the corresponding two non-pooled individual cumulus cell clusters of the two used oocytes from one donor. We calculated the t-test comparison between the two groups’ Z scores: the first group, red positive vs. blue negative; the second, red positive vs. yellow negative. The significant findings are depicted in Figure 3C–E. We found significance differences for 16 upstream regulators, of which five are classified as transcription regulators (SRF, Hdac, HTT, TP53, TGFBR2), three as ligand-dependent nuclear receptors (PPARA, NR3C1, ESR1), three as kinases (AKT1, INSR, TGFBR2), two as enzymes (KRAS and MGEA5), two as growth factors (TGFB1 and BMP2), and two as hormones (LH and FSH). These two hormones are directly related to follicular development and are among those used in ovarian stimulation. After corrections for multiple testing, 5 out of 16 remained significant: LH, FSH, HTT, INSR, and TP53. The one-to-one sample individual Z score comparison for FSH is also shown in Figure 3A. 

Moreover, the difference between the Z scores of the middle red-positive vs. upper blue-negative group on the one hand and the middle red-positive vs. lower yellow-negative group on the other revealed clear differences based on 14 significant canonical pathways, of which eight remained significant after correction for multiple testing (Figure 3D) as did seven disease and four bio functions categories (Figure 3E).

All 14 of the above-mentioned canonical pathways fell into the category of cellular signaling. Six involved signaling of cellular/organismal growth proliferation and development (PI3K/AKT, mTOR, ILK, integrin, ErbB4, and PCP); three were linked to apoptosis (induction of apoptosis by HIV1, LPS-stimulated MAPK signaling, and apoptosis signaling), and two were linked to cancer (PI3K/AKT and PCP) (Figure 3D). The above-mentioned categories clearly indicate the difference in proliferative phases of the two negative groups in comparison to the positive ones, and in comparison to each other they reflected the cumulus cell reaction towards hormonal stimulation. In addition, there were groups of signaling pathways that were part of either intracellular and second-messenger signaling (PI3K/AKT, signaling by Rho family GTPase, G alpha q, and integrin signaling) or cellular immune response (IL-8, leukocyte extravagation, production of NO, and reactive oxygen species in macrophage). In support of the different state of proliferation phase/apoptosis of both negative groups in comparison to the positive samples, the category of disease and function showed statistical significance in the category’s life cycle, cell death, and degeneration (Figure 3E).

### 2.8. Comparing our Expression Data with Available Published Datasets

As we have observed that little overlap existed between different studies concerning the molecular biomarkers that could be linked with the clinical phenotype, we searched for publicly available genome-wide expression data that were comparable to our present study (i.e., human, cumulus cells, long protocol treatment). Two available datasets were available and compatible with our study [26,42]. We aimed to perform comparisons on) the intersection between differentially expressed genes and on the degree of sample heterogeneity. 

The intersection between the differentially expressed genes was minimal among the three datasets when considered two at a time (33, 34, and 7 intersections between Borup et al. [42] and Demiray et al. [26], Demiray et al. [26] and the present study, and the present study and Borup et al. [42], respectively) and zero among all three together (Appendix A). The little intersections were also accompanied by the scrambling of phases of differentially expressed markers among the three data sets (i.e., what was upregulated or downregulated in one dataset was distributed in both up- and downregulation in another).

The degree of heterogeneity of the samples was estimated by calculating the intersection percentage in the results of 10 different tests between negative and positive samples after dropping a random 10% of the samples each time. We included the control data sets from male and female blood samples where differential expression between males and females was calculated (Appendix A). The global variance between different tests within one study was calculated and showed relative high variance in both the studies of Demiray et al. [26] (variance = 156.3) and Borup et al. [42] (variance = 143.9) followed by the present study’s (variance = 46.7). The blood control group showed a low variance of 15.9%. This, in our opinion, clearly showed the high heterogeneity of the cumulus samples from all studies and that ours was not particularly higher.

In addition, we analyzed the enrichment of IPA canonical analysis, disease and bio-function, and Tox function in the differentially expressed genes of the three sets. When we compared the predicted Z score, we found little consistency among all the studies: 5/22 for pathway analysis; 8/39 for disease and bio-function; and 2/5 for Tox analysis (Appendix A). 

## 3. Discussion

One of the aims of improving the clinical results of ICSI cycles is to predict the developmental potential of the transferred embryo, thus choosing the best for transfer and leading to less psychological stress for the affected couples and cost reductions for the health system. The use of biomarkers in cumulus cells surrounding the oocyte to predict the clinical outcome is a possible non-invasive approach. To this end, we performed molecular profiling, including, for the first time, simultaneous RNA and DNA methylation profiling on outer cumulus cells retrieved from ICSI cycles.

In order to obtain informative biomarker(s) without false positive influence of interfering factors, we determined, using literature-based knowledge and our experimentally obtained data, conditions that could potentially influence the results (i.e., choice of biomarkers). The main factors influencing the expression and methylation markers were inter-individual differences and differences in treatment protocol: short vs. long. 

The first could be a reflection on the genetic setup of individuals and their response to treatment. The inter-individual differences could be clearly observed (Figure 1A) in the unsupervised PCA samples and heatmap where the samples derived from the same individual, even for two different stimulations using the same protocol, were interconnected and clustered close together. This could be explained by genetic polymorphisms affecting general expression patterns or a response to hormone stimulation treatments. It is documented that genetic variations affect the global gene expression patterns [43,44] and that any response to induced hormonal treatment could be modulated by polymorphisms in the FSH receptor [40,45]. Therefore, an influence of the individual genetic make-up on the treatment outcome cannot be avoided. However, it could be predicted from previously known polymorphisms, where the treatment regimens had been precisely adopted for individual genetic setup. Unfortunately, in this study, we could not investigate whether the individual-specific SNPs were directly associated with cumulus expression profile, as patient DNA (from peripheral blood) was unavailable for this study. The second observed influence was the hormonal stimulation protocol, as the differences in the unsupervised classification of samples were clearly influenced by the protocol used, short or long, which reflected the use of different stimulating/treatment regimens (Appendix A). Therefore, we stratified the samples according to treatment protocol but could not do the same for the genomic/polymorphism because such information was not available. 

We identified individual single markers that were differentially expressed where NME6 and ASAP1 were under expressed in positive samples (i.e., continuing pregnancy). NME6 is the nucleoside diphosphate kinase 6, which specializes in the phosphorylation of nucleotide diphosphate, which regulates on cell growth. NME6 is ubiquitously expressed at low levels in most human tissues but is abundant in the kidney, prostate, ovary, intestine, and spleen [46]. It has also been found to be particularly abundant in zebrafish ovaries [47]. To date, it has not been experimentally proven whether NME6 is active in human cells. However, it seems to be localized in the mitochondria [48], suggesting a possible role in signaling by contributing to nucleoside metabolism and the ratio of different forms. The latter could itself regulate the growth of cumulus cells around the oocytes and probably the oocyte itself. ASAP1 stands for ArfGAP with SH3 Domain, Ankyrin Repeat and PH Domain 1. It plays a role in shaping the actin cytoskeleton and induces proliferation [49,50]. It is also overexpressed in many tumors and could trigger cellular movement and the ability of cells to move and thus promote metastasis [51,52]. As the process of oocyte maturation and ovulation necessitates cellular remodeling surrounding the oocyte it is not surprising that ASAP1 is considered one of the players indicating a proper maturation stage. However, we also saw a correlation of these two markers with endometrium size in the positive samples (Appendix A); therefore, it is questionable whether this is an effect of endometrium development with an effect on implantation that was also detected in cumulus cells or is a marker for oocyte development. It could be that endometrium development is the main determining factor to be considered and that it overwrites any oocyte factor. Future studies that group large samples based on endometrium size should be performed to determine if these two markers are endometrium or cumulus cell specific.

When comparing our gene marker results to previous results obtained from cumulus cell global expression, very little overlap could be identified (*p* < 0.05) (Appendix A). Assidi et al. (2011) [15] identified seven markers as the top differentially expressed between positive and negative samples (*NRP1*, *UBQLN1*, *PSMD6*, *DPP8, HIST1H4C*, *CALM1*, *TOM1*), most of which were reconfirmed by a second study by Assidi et al. (2015) [21], except for *HIST1H4C* and *TOM1*. Feuerstein et al. (2012) [17] identified three markers (*PLIN2*, *RGS2*, *ANG*), while Xu et al. (2015) [23] determined nine that were confirmed by RT-PCR (*BAI2*, *HDAC1*, *ELOVL5*, *ITGA6*, *RAD51*, *VCAN*, *PTGS2*, *POU5F1*, *VAMP7*) and Demiray et al. (2019) [26] identified five markers (*ZFP57*, *ARHGEF4*, *SOX8*, *FLJ39502*, *CNIH3*) according to clinical pregnancy and >2-fold change. Here, two observations must be highlighted: (1) among the above-mentioned top markers, there was no single marker overlap between two research sites, and (2) Assidi et al. clearly confirmed their initial study. To explain this, we proposed that laboratory-specific factors may affect results, by making single-gene markers non-transferable from one site to another. This could be related to (1) the cellular heterogeneity of isolated materials, (2) specific ethnicity of patient cohorts and genetic make-up, (3) the stimulation and triggering protocol of hormone treatment, (4) experimental procedures of isolating the tested materials, (5) data generation methods (i.e., RNA seq, microarrays or even RT-PCR), and (6) genetic heterogeneity of the patients. The three overlaps from the above-mentioned potential markers within our data are *TOM1* (log_FC_ = −0.28; *p* = 0.00157), *HDAC1* (log_FC_ = 0.35; *p* = 0.00239), and *PTGS2* (log_FC_ = −0.61; *p* = 0.0298).

Although no significant overlap in the single-marker genes from cumulus cells across different studies was observed, the use of gene expression as a biomarker for embryo quality was also questioned by others [53]. However, the increasing trend of using pathway analysis showed common affected pathways. Indeed, IPA and KEGG pathway analyses are able to show which pathways are controlled by differentially expressed genes and which distinguish positive from negative samples. In this respect, our data are in accordance with the general trend of considering apoptotic markers as predictive markers [15,26,54]. Our results also showed similarly affected pathways, such as cell death and survival in the category of bio function (for inter-individual differences see Figure 1C,D,F; for differences between positive and negative samples see Figure 2D,E,G). 

In this study, for the first time we were able to perform relative classifications among patient samples. This was possible because we analyzed a minimum of two individual non-pooled cumulus-cell cluster samples from each donor and took advantage of the IPA calculated differences in Z scores for a specific pathway or bio function. Therefore, we were able to classify on a linear scale our positive samples between two negative groups (Figure 3). Thus, the positive samples showed—in the middle of other signaling pathway—a response for cell death progression, apoptosis signaling, and several upstream regulator molecules. Here, identification of either FSH or LH as an upstream regulator was of particular interest (Figure 3C). Using this as biomarker, along with many others, we classified the samples based on predicted FSH activity/effect (described here as a response to FSH) on the expression profile in cumulus cells and classified the samples into three groups with the positive samples between the two negative groups. This suggests that one negative group showed a higher response to FSH, while the second group showed a lower one. In other words, too much FSH (or too long exposure) is as bad as too little (or too short exposure). We, therefore, suggest that as a first step the samples structure (different groups) should be determined and that the analysis should take this into consideration. This will increase the accuracy of biomarker discovery based on multiple group comparisons, not a simple differentiation between two groups (Figure 3B, Appendix A). 

The present study includes limitations and advantages. The former includes a relatively low number of patients as well as the absence of DNA polymorphism analysis that could be used to investigate/exclude genetic effects and help sort samples into subgroups. Our several advantages included the study of at least two individual cumulus complexes from each donor, which allowed Z score calculations to determine inter-patient variations, the simultaneous analysis of both gene expression and DNA methylation, and a correlation analysis of both expression and methylation markers with clinical parameters. 

## 4. Materials and Methods

### 4.1. Patient Samples and Clinical Parameters

For this study, eight couples (mean age, 33 years) undergoing ICSI treatment were recruited (Appendix A. All the individuals signed written consent for the study, which was approved by the ethics committee for clinical studies at the University of Bonn (approval number: 233/10). Two couples underwent repeated treatments separated by an interval. Therefore, we were able to collect samples from 10 different treatment cycles: VS1–VS13. There were three and seven sample groups collected after hormonal stimulation using the short (GnRH antagonist) and the long (GnRH agonist) protocol of ovarian stimulation, respectively (Appendix A). Individual cumulus clusters surrounding each oocyte were physically separated from the oocyte and further processed as non-pooled. Sample data on the spindle presence, Zona pellucida birefringence, pronuclear pattern, and estrogen level, endometrium size and lead follicle size were available for this study (Appendix A). Five cases resulted in a negative pregnancy test (hCG negatives), while in five cases, the test was positive. Of these, only three continued as normal pregnancies, whereas two aborted in the early stages.

### 4.2. Ovarian Hormone Stimulation Protocol and Outer Cumulus Cell Collections

Ovarian hormone stimulation was performed according to either the long or the short protocol (detailed in Appendix A). All patients participating in the study had regular cycles and normal ovarian hormones. The decision to use the long or short protocol was based exclusively on organizational grounds or the wish of the patient. Briefly, for the long protocol, the patients were administered 10 ng of GnRH agonist (Decapeptyl^®^ from Debiopharm) daily for about two weeks. After documentation of hypophyseal downregulation, ovarian stimulation was started with a daily dose of 150 IU rFSH (FINOX^®^ from Finox Biotech, Technikumstrasse 2 Burgdorf, CH-3401, Switzerland, Gonal F^®^ from Merck-Serono, Merck Serono GmbH, Alsfelder Str. 17, 64289 Darmstadt, Germany or Puregon^®^ from Merck Sharp & Dohme, Lindenplatz 1, 85540 Haar, Germany) for 11 to 14 days. For the short protocol, patients were administered 150 to 225 IU rFSH (from Merck or Puregon^®^ from Merck Sharp & Dohme or Bravelle^®^ from FERRING Pharmaceuticals, Ch. de la Vergognausaz 50, 1162 Saint-Prex, Switzerland) starting on day 2 of the menstrual cycle for 9–11 days; GnRH antagonist (Orgalutran^®^ from Merck Sharp & Dohme or Cetrotide^®^ from Merck-Serono) at a dose of 250 mg was given on days 6 or 7 until ovulation. The triggering/final maturation was performed by injecting 250 µg hCG (Ovitrelle^®^ from Merck, Frankfurter Str. 250, 64293, Darmstadt, Germany). Follicular puncture and cumulus–oocyte complex aspiration were performed about 36 hours post hCG injection.

Individual cumulus cell clusters were collected from oocytes immediately before the ICSI procedures. Upon the mechanical isolation of the outer cumulus cells without enzymatic treatment, the non-pooled individual outer-cumulus cells corresponding to each oocyte were directly stored at −80 °C in 70 µL of the buffer RLT Plus from Qiagen (cat. no.: 1053393, Hilden, Germany) until further use. 

### 4.3. Isolation of RNA and DNA

Messengers RNA (mRNA), small RNA, and DNA were simultaneously isolated using a combination of the following two kits: Qiagen All Prep DNA/RNA micro (cat. no. 80284, Hilden, Germany) and RNeasy Mini Kit (cat. no. 74104, Qiagen, Hilden, Germany). Briefly, the tubes with the collected outer cumulus cells were further filled to 350 µl of RLT Plus buffer and vortexed for one minute. The lysate was then loaded to an “All prep” DNA column and the bounded DNA was processed according to the kit instruction for DNA isolation. The flow-through containing the RNA was filled with 350 µl of 70% ethanol and loaded to an RNeasy mini-elute column and further processed according to kit instructions to isolate the long RNA. DNA and RNA were collected in 45 and 12 µL total volume and the yield was in the range of 200, 50, and 350 ng, respectively. 

### 4.4. Illumina Array-Based Analysis

We performed genome-wide expression and methylation analysis using the Illumina platform. Human methylation 450 DNA BeadChip were used for methylation analysis and human HT-12v4 expression BeadChip was used for expression analysis. The analysis was performed according to standard Illumina procedure except that the RNA was pre-amplified before use to ensure an optimum starting amount, using the TargetAmpTM-Nano labeling kit for Illumina Expression BeadChip (Epicenter, cat. no. TAN07924, Madison, WI, USA). 

### 4.5. Processing Methylation Arrays Data and Methylation Analysis

The raw Illumina 450K methylation microarray data (two-color IDAT files) were used to read into the R environment using the function “read.450K.exp”. The following normalization methods were used to analyze the methylation array data: subset-quantile within array normalization (SWAN) (Maksimovic et al., 2012 [55]), which performs normalization within the array and allows an Infinium I- and II-type probe on a single array to be normalized together. After normalization, the dataset was filtered for the detection of *p*-values > 0.01. After normalization and initial data processing as above, the data were passed into the Qlucore Omics Explorer 3.1 software (Qlucore, Ideon Science Park, Scheelevägen 17, 223 70 Lund, Sweden) for further analysis, visualization, and comparison against expression data. 

### 4.6. Processing of Expression Array Data and Differential Expression Analysis

Raw IDAT files were obtained from the Illumina human HT-12v4 arrays. These raw IDAT files were entered into Genome Studio to convert them into readable text files. Processing of raw text files was carried out in R using the lumi package. (1) LumiR function was used to read raw .txt data. (2) Data were background corrected, normalized, and stabilized using the lumiExpresso function. Normalization was performed using quantile normalization, whereby variance was stabilized using the variance stabilization transformation (VST) method. (3) Processed data were further filtered for detection of *p*-values. Samples that were not expressed in comparison to background noise were filtered out of the processed data. Final processed data were entered into Qlucore Omics Explorer for differential expression analysis. A *t*-test was used for a two-sample comparison, while ANOVA was used for multiple-sample comparisons. A threshold of *p* < 0.05 or 5% of a false discovery rate (FDR) was considered statistically significant for both comparisons.

### 4.7. Gene Ontology and Pathway Analysis 

Gene ontology analysis was carried out using the “Bingo” plugin in Cytoscape 3. Biological process was identified at 5% of FDR for the differentially expressed genes. The “EnrichmentMap” plugin in Cytoscape 3 was used for the visualization of gene ontology. Pathway analysis and upstream regulator analysis were performed using Ingenuity Pathway Analysis (IPA) from Qiagen. Regarding plot and visualization, heatmaps and principal component analysis (PCA) plots were created in Qlucore Omics Explorer. Volcano plots and other statistical plots were visualized using R, Graphpad prism or Cytoscape 3.

## 5. Conclusions

In conclusion, this study showed that experimental and clinical parameters and the heterogeneity of samples affect expression profiling in cumulus cells. Therefore, to determine reliable biomarkers associated with a positive pregnancy, variable results that could be introduced by experimental and clinical parameters should be excluded by stratifying the samples. For this, we recommend using ontology and results of pathway analysis involving upstream regulators as ubiquitous universal markers rather than the expression of single-gene markers. Finally, we recommend that future studies perform single cell analysis on cumulus cell complexes to solve the heterogeneity of cells in the cumulus complex.

## Figures and Tables

**Figure 1 ijms-22-06377-f001:**
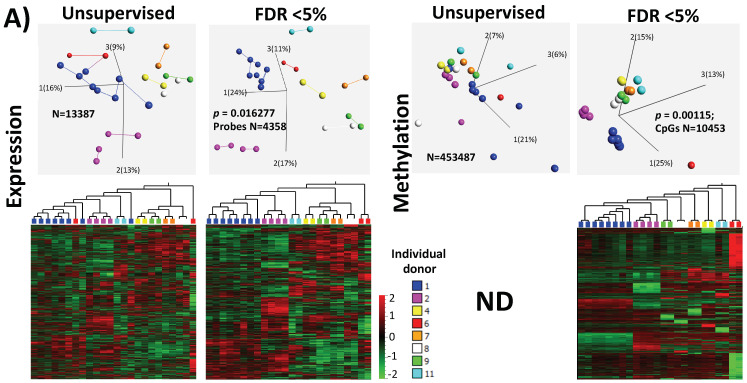
Inter-individual differences in expression and methylation. (**A**) 3D-PCA (upper part) and heatmaps (lower part) representing expression and CpG methylation profiling of 24 individual cumulus cells derived from eight different women. Left and right parts represent expression and methylation, respectively, unsupervised and ANOVA for multiple comparisons at FDR < 5% analysis are shown. (**B**) Correlation between expression and methylation. Right upper part shows Venn diagram of intersection/overlap of DEG and DMC; in total 659 genes overlapped with 1043 investigated CpGs. The correlation of the overlaps is shown at the left upper part as −log_10_ (*p*-values) vs. the relative position to the transcription start site. In addition, the representative density/fitting curve is shown as a dotted line (red or blue represent negative and positive correlation, respectively). The lower panel shows correlation graphs of the highest 14 correlations (three positive and 11 negative) (The color of the dots represents the individual treatment, and only transferred samples were used in this analysis). (**C**) Ontology analysis of the group of genes showing strong inter-individual differences (ANOVA analysis at FDR < 0.5); both biological process and cellular component are shown. The vertical arrow indicates the TSS; others arrows are linking to the dots with the number(s). (**D**) Top-10 affected canonical pathways as determined by IPA. The lower *x*-axis represents the −log(B–H *p*-value) for enrichment, while the upper *x*-axis represents the percentages of variable genes in a given pathway. (**E**) Top-10 upstream regulators (predicted by IPA) are listed with the overlapping *p*-values and numbers of regulated genes in the data; FSH regulated proteins are represented with symbols reflecting their functions and in their subcellular localization. (**F**) The top categories in disease and biofunction (predicted by IPA). The top-10 significant subcategories according to *p*-values are listed below together with the *p*-values.

**Figure 2 ijms-22-06377-f002:**
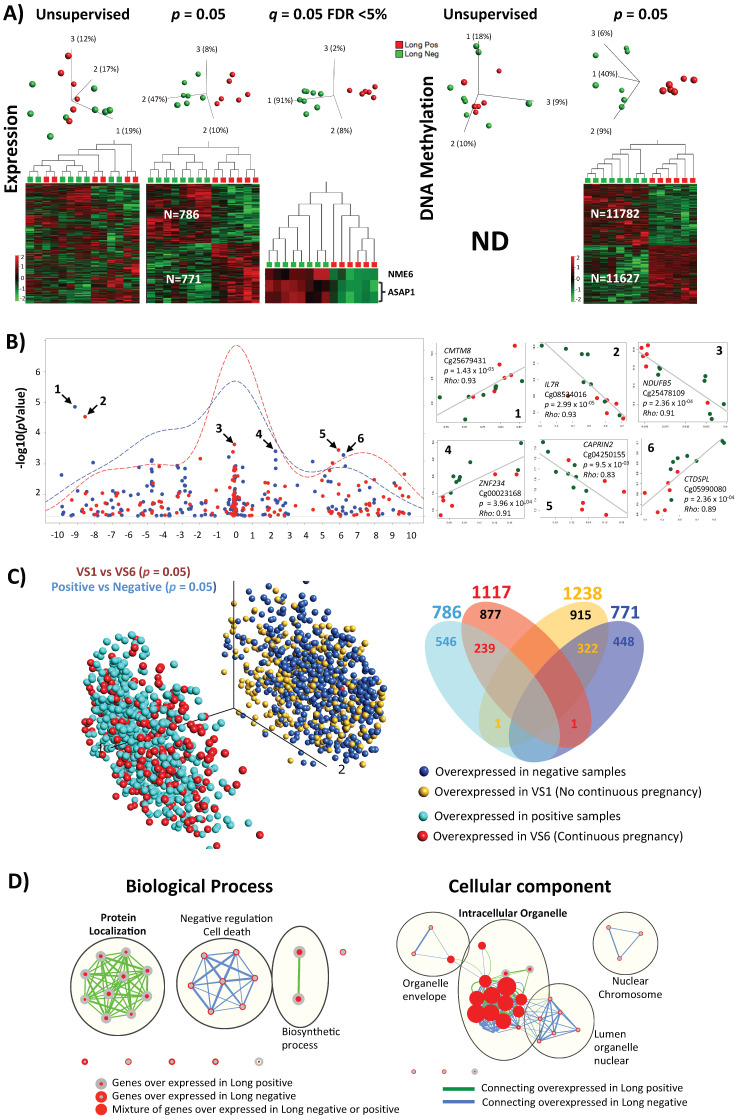
Differences in expression and CpG methylation profiles between pregnant positive and pregnant negative samples of the long protocol. (**A**) 3D-PCA plots and heatmaps of expression and CpGs methylation profiling. Result of unsupervised and significance at *p* = 0.05 and FDR < 5%, when applicable, are shown. (**B**) Correlation between differentially expression genes and differentially methylated CpGs. The correlation of the two data overlaps is shown on the left as −log_10_ (*p*-values) vs. the relative position to transcription start site (TSS = 0; every unit is 1 Kb), representative density/fitting curve is also shown in dotted line (red or blue represents negative or positive correlation, respectively). The right panel shows individual data for the highest six correlations (three positive and three negative correlations) (the color of the dots represents the pregnancy test result with red (positive) and green (negative)). (**C**) Comparison between results of successive analyses of two treatments of the same individual (VS1 negative vs. VS6 positive; shown in Figure 2) on one side and comparison between all pregnancy negative and pregnancy positive samples on the other (part A above). The left part represents the 3D variables of the PCA expression data (part A above), where the overlap of the overexpressed and the underexpressed genes between the two comparisons is in a near-complete phase. The right part represents a Venn diagram showing the overlap of the differentially expressed genes for the two comparisons in question. (**D**) Ontology analysis of the genes with difference in expression at *p* = 0.05 for biological process and cellular component are shown. IPA analysis (at *p* < 0.05%) for top-10 affected (**E**) canonical pathway, (**F**) upstream regulators (details of FSH-affected targets are shown on the left), and (**G**) diseases and biofunction.

**Figure 3 ijms-22-06377-f003:**
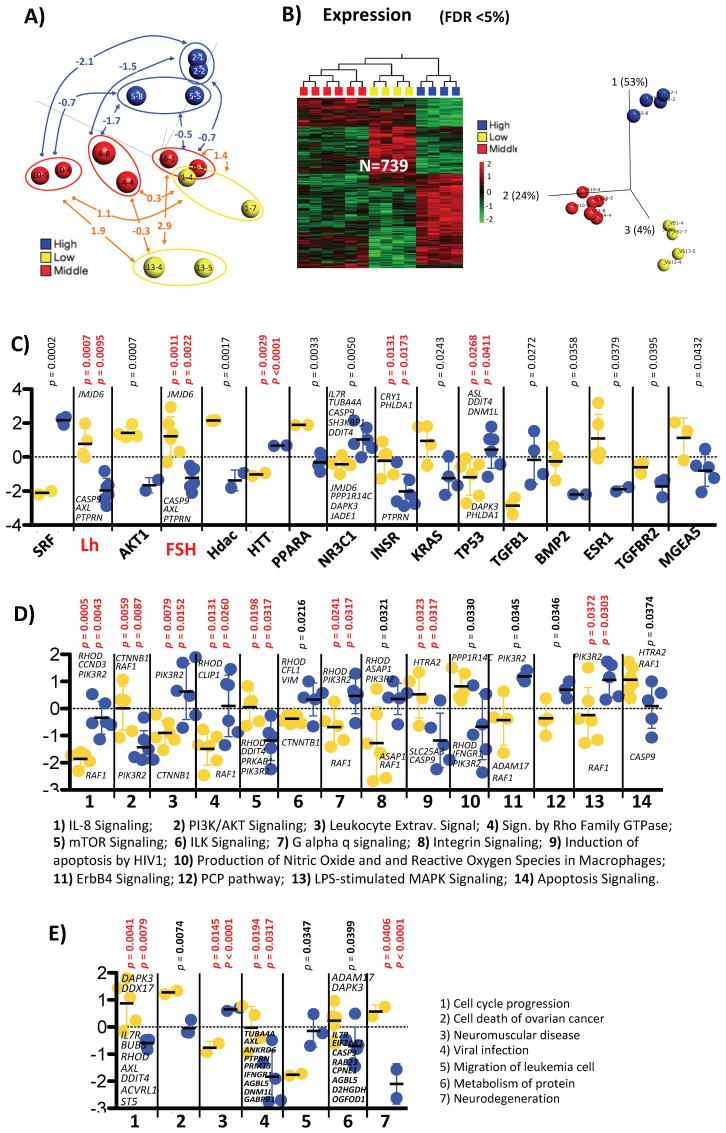
Detailed analysis of differentially expressed genes between positive and negative pregnancy test in the long protocol. (**A**) Unsupervised sample 3D-PCA plot (shown also in Figure 2A) overlain with differences in FSH Z score between each of the pregnancy positive samples and each of the pregnancy negative samples. Blue, red, and yellow balls correspond to the samples with high, middle (pregnancy positive samples), and low FSH Z scores. (**B**) 3D-PCA and heatmap plots based on ANOVA (FDR < 5%) of the three sample groups classified by FSH scores. The three groups are effectively separated based on 739 expression probes and 88 CpGs sites corresponding to 678 genes and 63 genes, respectively (with three gene overlaps: *AFAP1, GPBP1L1*, and *LIMS2*). Comparisons of Z scores between middle-low and middle-high FSH levels for upstream regulators (**C**), canonical pathways (**D**), and disease and biofunction (**E**). Blue- and yellow-filled circles correspond to Z score differences between each of the middle FSH level group with each of the high-FSH and low-FSH level group, respectively. *t*-test *p*-values are shown above each scatter plot. Additionally, the significant *p*-value after multiple corrections is shown in red. The overexpressed gene names contributing to the significant differences are also shown in the box.

## Data Availability

All the data has been submitted to the gene expression omnibus (GEO) under accession number GSE144665.

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
