# Peer review of "Stratifying Cumulus Cell Samples Based on Molecular Profiling to Help Resolve Biomarker Discrepancies and to Predict Oocyte Developmental Competence"

_ijms, 2021, doi:10.3390/ijms22126377_

Round 1

Reviewer 1 Report

  1. The authors showed hard work in the study. The manuscript is informative and well written. But some word misspellings are found.
  2. The term of short protocol may be misused. GnRH antagonist protocol is more precisely term in the study.
  3. The resolution of the figures is too low to seeing clearly.
  4. In Figure 1E, the picture is not correct. FSH is an extracellular hormone, which is combined with its receptor on cellular membrane. However, the picture showed FSH is located at intracellular site.
  5. It is a little pity that positive and negative pregnancy in GnRH antagonist treatment are not grouping and comparison in the study.

Author Response

  1. The authors showed hard work in the study. The manuscript is informative and well written. But some word misspellings are found.

Answer: We addressed this issue by proofreading the manuscript.

  1. The term of the short protocol may be misused. GnRH antagonist protocol is a more precise term in the study.

Answer: we have added on page 3 on the first appearance of the short protocol terms GnRH antagonist between parenthesis. However, for simplicity and ease of reading, we keep using the terms short and long protocols throughout the manuscript.

  1. The resolution of the figures is too low to seeing clearly.

Answer: We have updated the figures to increase their resolutions wherever needed.

  1. In Figure 1E, the picture is not correct. FSH is an extracellular hormone, which is combined with its receptor on the cellular membrane. However, the picture showed FSH is located at an intracellular site.

Answer: We are sorry for this mistake caused by the picture production. What is meant is the localization of interacting molecules (outer cell, membrane, cytoplasm, or nucleus) and FSH is meant to be above the cell. To remove confusion we wrote below FSH: extracellular.

  1. It is a little pity that positive and negative pregnancy in GnRH antagonist treatment is not grouping and comparison in the study.

Answer: We also feel the same. Especially that in the group of samples we don’t have GnRH antagonist positive samples.

Reviewer 2 Report

This manuscript covers an interesting topic - predictive biomarkers for selection of the most suitable germ cells for fertilization and the best embryo for intra uterine transfer. They found that the presence of several parameters affects analysis outcome, such as inter-individual differences, inter-treatment differences and above all - treatment protocol specific differences. Comparing the pregnancy outcome indicated single gene markers (NME6 and ASAP1, FDR <5%) upstream regulators (e.g. EIF2AK3, FSH, ATF4, MKNK1, TP53) and several bio-functions related to cell death/apoptosis, cellular growth and proliferation. Authors suggested the use of pathway analysis as a more reliable and universal marker identifier method, to predict oocyte developmental potential.

There are minor issues which are needed to be considered,

  1. Title is too long and confusing,
  2. Introduction, method and results are well written with comprehensive information.
  3. In fig2.A, is the q value is 0.05?
  4. In the first sentence of discussion, “O One of the aims to improve” needs to be corrected,
  5. Discussion section included detailed discussion which covers the aim of the study.

Author Response

  1. The title is too long and confusing,

Answer: We modified the title to read as follows: Stratifying Cumulus Cells Samples Based on Molecular Profiling Helps Resolve Biomarkers Discrepancies to Predict Oocyte Developmental Competence.

  1. Introduction, method, and results are well written with comprehensive information.

Answer: thank you for your comment.

  1. In fig2.A, is the q value is 0.05?

Answer: in figure 2A as the reviewer mentioned the q value is 0.05 for the third expression figure (from left to right), as for the methylation markers there is no significance at q=0.05.

  1. In the first sentence of discussion, “O One of the aims to improve” needs to be corrected,

Answer: We apologize for this mistake the ‘O’ is removed.

  1. The discussion section included a detailed discussion that covers the aim of the study.

Answer: We are thankful for the comment.